# Mature Triphyllic Cystic Teratoma of the Posterior Mediastinum in a Fetus: A Case Report and Literature Review

**DOI:** 10.3390/diseases11040144

**Published:** 2023-10-18

**Authors:** Cecilia Salzillo, Andrea Quaranta, Giovanni De Biasi, Gerardo Cazzato, Gabriella Serio, Antonella Vimercati, Pierpaolo Nicolì, Leonardo Resta, Andrea Marzullo

**Affiliations:** 1Department of Precision and Regenerative Medicine and Ionian Area (DiMePRe-J), Section of Pathology, University of Bari, Piazza Giulio Cesare 11, 70121 Bari, Italy; giovanni.debiasi@uniba.it (G.D.B.); gerardo.cazzato@uniba.it (G.C.); gabriella.serio1@uniba.it (G.S.); leonardo.resta@uniba.it (L.R.); andrea.marzullo@uniba.it (A.M.); 2School of Medicine, University of Bari, Piazza Giulio Cesare 11, 70121 Bari, Italy; a.quaranta35@studenti.uniba.it; 3Department of Interdisciplinary Medicine (DIM), Unit of Obstetrics and Gynecology, University of Bari, Piazza Giulio Cesare n.11, 70121 Bari, Italy; antonella.vimercati@uniba.it (A.V.); pierpaolonicoli@icloud.com (P.N.)

**Keywords:** mature triphyllic cystic teratoma, posterior mediastinum, fetal teratoma, germ cell tumor, prenatal diagnosis

## Abstract

A teratoma is a neoplasm composed of cell populations or tissues that are reminiscent, in their appearance, of normal elements derived from at least two embryonic layers. Fetal mature teratomas are normally benign, cystic, and typically occur along the midline, while they are rare in the posterior mediastinum. Teratomas are frequently solitary; however, they may sometimes be associated with other congenital anomalies and/or with chromosomal abnormalities. Clinically, they are often asymptomatic but can occasionally cause compression symptoms. Prenatal diagnoses are uncommon and made with ultrasonography; differential diagnosis with other congenital conditions is mandatory. We report the case of a 21 weeks of gestational age old fetus with a mature triphyllic fetal cystic teratoma, grade 0, located in the right posterior mediastinum. The tumor presented as a 3 cm wide cystic mass that caused a contralateral shift of the surrounding structures. Histological examination later revealed the presence of derivatives of the three germ layers, such as hyaline cartilage, smooth muscle, nervous tissue, and a respiratory-type epithelium.

## 1. Introduction

The term teratoma comes from the Greek “(teras, terátos)” meaning “monster” and the suffix “-(-oma)”, suggesting a process or an action [1,2]. Teratomas are germ cell tumors that represent derivatives of two or more embryonic layers (ectoderm, mesoderm, and endoderm) [1,2,3].

Etiopathogenesis remains unclear, although three main theories are described: An origin from totipotent cells left behind in their migration through Hensen’s node, explaining the frequent median and paraxial distribution of teratomas;Teratomas explained as a consequence of an incomplete twinning, which could clarify intracranial, mediastinal, abdominal, and sacrococcygeal localizations;An origin from primitive germ cells migrating along the midline from the hindgut to the crista gonadalis which explains the pelvic and gonadal distribution of some teratomas [2,4].

Teratomas are the most common congenital tumors, representing 16.6% of all fetal neoplasms, with an incidence of 1:20,000 to 1:40,000 live births [5]; nonetheless, another single-center study showed a significantly higher incidence, of 1.3:1000 pregnancies, with a female to male ratio of over 2:1 [6].

Teratomas are usually located along the midline or in paramedian position and, anatomically, can be either gonadal (20%) or extragonadal (80%). 

The most frequent extragonadal site in infants is the sacrococcygeal area (40%), followed by cervical and retroperitoneal distributions. That said, several other locations, such as intracranial/cerebral [7], mediastinal [5,8,9], gonadal, pharyngeal, hepatic, cardiac, pleural, and subcutaneous, have been reported as less common.

Mediastinal teratomas are among the least frequent, with reported percentages ranging from 2.6 to 4 in the literature [5,8].

Past analyses show that teratomas are frequently solitary, though they might sometimes be associated with other malformations and chromosomal abnormalities, such as Klinefelter syndrome [5,10].

Prenatal diagnosis is critical as teratomas are often asymptomatic; however, due to their large volume, they can occasionally compress the lung parenchyma, causing lung hypoplasia and severe respiratory failure at birth, or compress the heart, causing nonimmune fetal hydrops (NIHF) and death [5,9].

As for the incidence of conatal teratomas, a recent single-center study by Simonini et al. reported only seventy-nine cases in seventeen years [10]. 

Diagnosis is usually made during routine pregnancy checkups by ultrasonography, and differential diagnosis with other mediastinal masses is crucial, such as bronchogenic cysts, congenital cystic adenomatoid malformations of the lungs, diaphragmatic hernias, bronchopulmonary sequestrations, or thymic cysts [5,8,9].

Treatment varies in accordance with the clinical presentation and the patient’s age.

To this day, while in the literature several reports of fetal anterior mediastinal teratomas can be found [5,8,9], there are only few reports of posterior mediastinal teratomas, none of which were diagnosed in a fetus or prenatally [11,12,13].

## 2. Materials and Methods

### Immunohistochemistry

Immunohistochemical analysis was performed on 4 μm thick FFPE (formalin-fixed, paraffin-embedded) tissue sections, using 3,3′-Diaminobenzidine as a chromogen. 

Cytokeratin 7 detection was performed with clone RN7 (Leica Biosystems, Wetzlar, Germany) to highlight respiratory-type epithelium, which shows diffuse expression of this protein and no expression of Cytokeratin 20 (a marker for other endodermic derivatives, such as gastric and intestinal epithelium). Cytokeratin 20 staining was performed using clone KS20.8 (Leica Biosystems, Wetzlar, Germany). Thyroid Transcription Factor 1 (TTF1) detection was performed using the SPT24 antibody clone (Leica Biosystems, Wetzlar, Germany) at a 1:100 dilution to identify any Clara cells and/or developing type II pneumocytes. Staining for Gliofibrillary Acid Protein (GFAP) was performed with Clone GA5 (Leica Biosystems, Wetzlar, Germany) to point out any presence of astrocytes, due to the vast areas of possibly glial tissue made of small, star-like cells. Non-nervous tissue was used as an internal negative control. External positive controls were used for all the reactions as recommended by the data sheet. Smooth muscle actin (SMA) staining was performed using clone 1A4 (Dako Agilent, Santa Clara, CA, USA) to confirm the presence of smooth muscle cell bundles. For this purpose, the wall of small vessels was used as internal positive control. 

Analysis was performed on a multiheaded brightfield microscope by G.S., A.M., C.S., G.D., and A.Q.

## 3. Case Report

At 23 weeks of gestation, a Pakistani 32-year-old female with gravida 1 parade 0 abortion 0 (G1P0Ab0) was referred to the Prenatal Diagnosis Unit of the secondary University Hospital of Bari (Southern Italy) for the detection of a suspicious thoracic cyst during a routine second-trimester fetal anatomy scan. 

It was very difficult to obtain the patient’s full history due to the language barrier as an interpreter was not available; however, as far as the physicians understood, the patient had a healthy first child, had conceived spontaneously, and had no significant medical history. She underwent the first trimester ultrasound, but not the combined test.

Firstly, the operators redated the pregnancy based on the crown–rump length (CRL) detected during the first trimester ultrasonography (−7 days); so, the patient was at 21.5 weeks of gestation according to ultrasound. Despite the sonographic redating, the ultrasound evaluation (performed with a GE Voluson E10 Ultrasound Machine) showed that the fetal biometry was lower than the first percentile for gestational age as per the Williams package, confirming the presence of a severe early-onset intrauterine growth restriction (IUGR). Amniotic fluid was increased (mild polyhydramnios, with a 12 cm maximum vertical pocket of amniotic fluid). The color Doppler examination revealed normal blood flow in all fetal districts; instead, the pulsatility index (PI) of both uterine arteries was higher than the 95th percentile for gestation age, with a second-grade bilateral notch.

The morphological examination revealed the presence of a large anechoic cystic mass with unregular borders occupying the entire right hemithorax (LD = 2.7 cm; APD = 2.5 cm; TD = 2.4 cm), which apparently originated from the lung parenchyma (Figure 1A). The cyst had no internal or perilesional vascularization. Fetal heart appeared to be anatomically normal, but it was extremely shifted on the left, together with the mediastinum, due to the compression caused by the cyst. The remaining fetal anatomy seemed to be regular, except for the presence of marked ascites (Figure 1B). 

The ultrasound finding was primarily suggestive of a right large bronchogenic cyst, to include in differential diagnosis with other thoracic cystic formations, from more common anomalies like congenital cystic adenomatoid malformations (CCAMs) to rarer ones such as mediastinal teratomas.

A multidisciplinary team, including obstetricians, neonatologists, and pediatric surgeons, extensively informed the parents about this rare condition. Clinicians explained to the couple that the association between the thoracic cystic formation was complicated by ascites and polyhydramnios, and the severe early-onset fetal growth restriction was suggestive for poor prognosis. They informed the parents about all the possible management alternatives (from fetal tumor cyst fluid aspiration and observation to termination of pregnancy (TOP)) and suggested a complete genetic screening by late amniocentesis, fetal magnetic resonance imaging (MRI), and serial ultrasound checks. The couple eventually refused further investigations and opted for elective TOP. 

The woman was hospitalized and placed into the “red zone” of the Obstetrics Unit, due to a positive SARS-CoV-2 admission test (despite being the patient vaccinated). The abortive labor was induced by administration of prostaglandins and the woman gave birth to a stillborn. After delivery, she underwent surgical removal of the placenta. The fetus and relative placenta were sent to the Pathology Unit for a routine histological examination.

Upon external examination, the fetus was characterized by a marked abdominal distension. 

Internal examination revealed an accumulation of yellowish fluid in the abdomen and a large retropulmonary and supradiaphragmatic, 3 cm wide, paramedian cystic neoformation, prevalently situated on the right side of the posterior mediastinum.

The cyst adhered to the aorta and diaphragm, did not show any continuity with the respiratory or gastrointestinal tract, and caused a contralateral dislocation of the surrounding anatomical structures (Figure 2a).

Said formation was single, spherical in shape, with a smooth surface, purplish white in color and, after cutting it, it appeared to be unilocular and with a serohematic content (Figure 2b).

The remaining organs and the placenta appeared macroscopically unremarkable.

Histological examination of the neoformation revealed a large cyst bordered by a ciliated respiratory-type epithelium surrounded by bundles of smooth muscle cells (Figure 3a), with seromucinous glands, hyaline cartilage, and nervous tissue (Figure 3b).

Immunohistochemistry showed that the ciliated respiratory-type epithelium stained positively for CK7 (Cytokeratin 7) (Figure 4a) and was completely negative for TTF1 (Thyroid Transcription Factor 1) and CK20 (Cytokeratin 20); while the smooth muscle bundles were strongly and diffusely positive for SMA (smooth muscle actin) (Figure 4b) and nervous tissue was clearly positive for GFAP (Gliofibrillary Acid Protein) (Figure 4c).

Microscopically, the placenta was characterized by hydropic villi with an irregular profile from primary damage of the embryo. Microscopic examination of the remaining organs showed morphological features coherent with gestational age.

## 4. Results

Histologically, the cystic neoformation was characterized by respiratory epithelium with seromucinous glands, surrounded by a smooth muscle layers, hyaline cartilage, and nervous tissue. No immature tissue was detected in all the specimens examined. These features were also confirmed by immunohistochemistry. All said aspects guide us towards the diagnosis of a mature triphyllic fetal cystic teratoma, grade 0.

## 5. Discussion

A teratoma is a neoplasm composed of various cell types or tissues that are reminiscent, with their appearance, of normal elements derived from more than one of the three germ layers; even so, in the adult, ovarian monophyllic teratomas (teratomas that derive from only one germ layer) have been documented. A remarkable example is struma ovarii, an abnormal growth of fully mature thyroid tissue, often with endocrine activity, and capable of causing hyperthyroidism [14].

Histologically, teratomas can comprise tissues deriving from each of the three germ layers; among those, ectodermal components, such as nervous tissue, usually neuroglia, and skin, habitually with hair and nails, are regularly observed in congenital masses; on the other hand, endodermal components are uncommonly seen, yet often perfectly differentiated. Typical mesodermal findings are cartilage, bone, teeth, and muscle [2,15].

Specifically, a mature teratoma is identified by the sole presence of well-differentiated tissues, while an immature teratoma is defined when at least one type of undifferentiated tissue is present [2,15].

A mature teratoma is a benign neoplasm, in most cases cystic, composed of tissues derived from the three embryonic layers. This is the most frequent type of teratoma, and its diagnosis is confirmed only after surgical excision by histologic examination [1].

Nevertheless, distinctive features of immature teratomas are neuroectodermal elements, e.g., ependymal rosettes or other aggregations of neuroepithelial tissue, that at times show atypia, a high mitotic count, or hypercellularity, and are often intercalated with well-differentiated areas [5].

All teratomas are capable of autonomous growth, although only immature ones may exhibit malignant or recurrent behavior.

Grade in teratomas of the adult ovary plays a key prognostic role [16,17]. It is defined by observing the presence and features of immature nervous tissue using either a dualistic approach (low grade or high grade) [17], or a four-level grading system, known by some authors as Norris’ grade, which ranges from grade 0, indicating a fully mature teratoma, to grade 3, indicating a highly immature tumor [17]. Grades 1–3 imply the presence of a percentage of immature neuroepithelium ranging from <10 to >50% [2]. 

According to these criteria, our case corresponds to a grade 0 (fully differentiated, mature) teratoma. 

However, research shows that the benefit in grading congenital or fetal teratomas like adult ones is arguable and that grade based on the presence of immature elements is hardly predictive of recurrence or malignancy [16].

Furthermore, immature nervous tissue in a fetal or infantile teratoma is not necessarily a negative prognostic feature, while focal presence of yolk sac tumor in the context of an immature teratoma, both in adults and in fetuses or small children, has been reported [16,18] and has been proven to be a key predictive trait for recurrence in infants.

A higher expression of p53 has been demonstrated to be a common characteristic of the more aggressive immature teratomas [19].

An analysis of p53, as well as Ki67 proliferation index, was not performed on our specimen, since mature teratomas are fully benign lesions, and malignant behavior and aggressiveness in teratoma are evaluated on the basis of the presence of immature tissues [14]. 

As previously mentioned, various sources define the mediastinum as a rather uncommon location for fetal or congenital teratomas. Mediastinal teratomas are more commonly extracardiac than intracardiac: primitive intracardiac teratomas are extremely rare; reports of a dozen cases can be found in the literature, and they are said to represent approximately 1–5% of all pediatric neoplasms. Intracardiac location is especially unusual [20].

Histologic examination may unveil solid or cystic aspects and mature or immature components from the three germ layers with or without calcification [5,9].

Prenatal diagnosis plays a central role; ultrasonography is the main method of prenatal evaluation for fetal malformations and thoracic masses [21].

A cystic and solid mass in the anterosuperior mediastinum, polyhydramnios, and hydrops fetalis are the typical ultrasound features to be found in fetuses with mediastinal teratomas.

Depending on their site and size, teratomas, like all kinds of thoracic masses, can cause an esophageal, pulmonary, and cardiac compression syndrome, occasionally responsible for intrauterine fetal demise. 

Esophageal compression causes a decrease in swallowing of amniotic fluid with a subsequent polyhydramnios, while hydrops is due to the tumor obstructing venous return from the placenta. Appearance of fetal hydrops is considered a critical sign, and, according to previous authors, is a predictive sign of imminent fetal death [2].

An appropriate differential diagnosis of cystic congenital thoracic lesions can be extremely challenging. Bronchogenic cysts, CCAMs of the lungs, diaphragmatic hernias, bronchopulmonary sequestrations, or thymic cysts must be considered first [8].

On ultrasonography, a thoracic teratoma usually appears as a cystic and solid mass (in variable percentages depending on the histological composition of the tumor), possibly containing calcification spots showing an acoustic shadow cone [22]. In our case, the ultrasound presentation was atypical, and a bronchogenic cyst was firstly suspected because of the round, homogeneous, fluid-filled, unilocular mass aspect with irregular borders. In addition to the unusual sonographic appearance, ultrasound differential diagnosis was particularly problematic due to the large size of the cyst, which prevented sonographers from understanding which part of the mediastinum it originated from. 

According to some authors [23], fetal magnetic resonance imaging (MRI) can provide more information for a precise diagnosis, due to the better identification of fat, sebum, and adipose tissue, which are very suggestive for a teratoma.

Others [24] state that prenatal MRI usually provides little additional diagnostic information that would change its management, and its routine application is therefore probably not cost-effective. 

The role of amniocentesis in the management of fetal lung lesions remains unclear. Since these are not known to be intimately related to genetic disorders, an amniocentesis is considered unnecessary when a solitary lung lesion is identified. However, if a concomitant anomaly is discovered, such as the severe early-onset IUGR in our case, complete genetic screening should be performed [25].

The prenatal management of fetal thoracic masses depends on the diagnostic suspect and, mostly, on the estimated prognosis for fetus. It is widely accepted that the overall perinatal prognosis of a fetus affected by a suspicious lung malformation is more strongly associated with its overall growth during pregnancy than with its underlying histology. Numerous studies have demonstrated that mass size, rather than the histological type of the lesion, is more predictive of perinatal outcome based on a variety of outcome measures, including fetal hydrops, neonatal respiratory symptoms, and likelihood of lung resection in the newborn period [23,24,25]. The most common system of measurement for lesion size is the CPAM volume ratio (CVR), which measures the ratio between the three-dimensional size of the lesion and fetal head circumference. According to recent data, the maximum CVR during pregnancy is believed to be the best predictor of perinatal outcome in fetuses with lung malformations [22]. The CVR formula is
$$\mathrm{CVR}=\frac{width\;\left(cm\right)\times depth\;\left(cm\right)\times length\;(cm)\times 0.52}{head\;circumference\;}$$

A large, multi-institutional study by the Midwest Pediatric Surgery Consortium [24] has clearly shown that an initial CVR ≤ 1.4 identifies fetuses at very low risk for hydrops, and a maximum CVR < 0.9 is associated with asymptomatic disease at birth. The application of this concept on our case report (fetal head circumference = 16.2 cm; CVR = 0.52) revealed a favorable ratio. However, in our case, the association between a fetal thoracic cyst complicated by ascites and mild polyhydramnios and an early-onset IUGR (with possible genetic disorders) worsened the prognosis. Unfortunately, the couple refused the invasive investigation we proposed, and the cytogenetic examination could not be performed.

Upon macroscopic examination, the fetus already showed signs of hydrops caused by the presence of a cystic neoformation in the right posterior mediastinum, responsible for the displacement of the contralateral organs and a compression syndrome. Thus, already macroscopically, this was clearly not a cCAML.

Histologically, the cystic neoformation was characterized by respiratory epithelium surrounded by smooth muscle with seromucinous glands, hyaline cartilage, and nervous tissue. These features were also confirmed by immunohistochemistry. All said aspects guide us towards a mature triphyllic fetal cystic teratoma, grade 0.

## 6. Conclusions

In the presented case, the macroscopic, histopathological, and immunohistochemical tests are conclusive for the diagnosis of mature triphyllic cystic teratoma, grade 0, in a male fetus with morphological features corresponding to the 21st week of gestational age, predominantly located in the right posterior mediastinum. The etiopathogenesis behind this finding is unclear, although one could speculate that it is the result of an abnormal migration of germ cells transferring from the hindgut to the crista gonadalis along the midline (theory 2 from the introduction). An incomplete twinning process (theory 3) cannot be excluded. A prenatal multidisciplinary team, including obstetricians, pediatrics, and pediatric surgeons should inform the parents about the fetal prognosis, mostly based on the initial size of the mass, its relative growth, and all the management options, in order to give them the chance to freely and consciously choose the fate of their pregnancy.

## Figures and Tables

**Figure 1 diseases-11-00144-f001:**
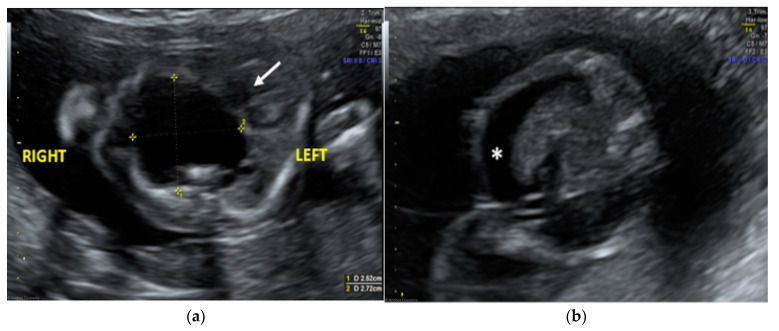
Transabdominal ultrasound managed at 23 weeks of gestational age according to amenorrhea (21.5 weeks according to first ultrasound): (**a**) Transverse fetal thoracic scan showing the large anechoic cystic formation occupying the entire right hemithorax. Fetal heart (arrow) and mediastinum were extremely shifted on the left due to the cyst compression; (**b**) transverse fetal transabdominal scan showing marked ascites (*).

**Figure 2 diseases-11-00144-f002:**
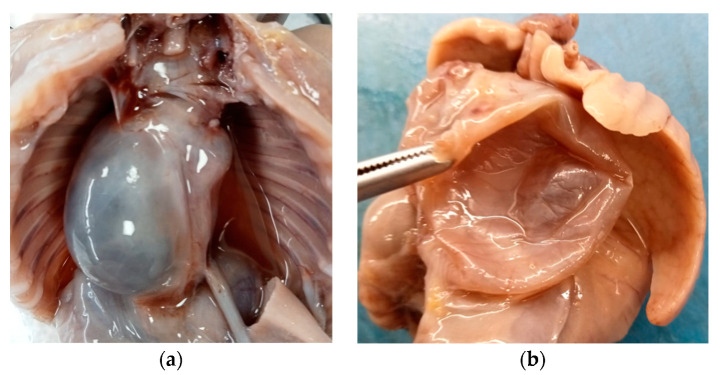
Mature triphyllic fetal cystic teratoma: (**a**) Cystic formation on the right side, retropulmonary and supradiaphragmatic, adhering to the aorta and diaphragm, with no continuity to other anatomical structures. (**b**) The cyst was single, spherical in shape, with a smooth surface, purplish white in color and, after cutting it, unilocular with a serohematic content.

**Figure 3 diseases-11-00144-f003:**
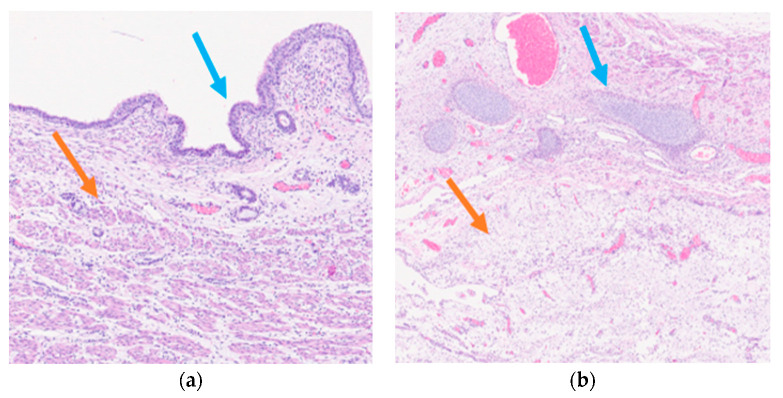
Mature triphyllic fetal cystic teratoma: (**a**) EH 5×: Ciliated respiratory epithelium (blue arrow) surrounded by bundles of smooth muscle (orange arrow); (**b**) EH 2.5×: Hyaline cartilage (blue arrow) and nervous tissue (orange arrow).

**Figure 4 diseases-11-00144-f004:**
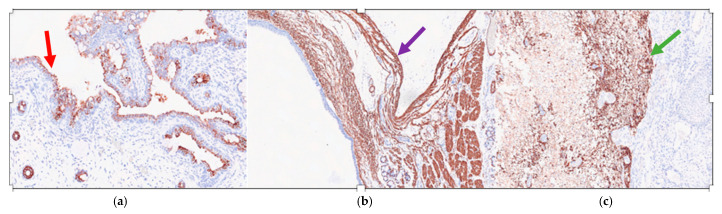
Mature triphyllic fetal cystic teratoma: (**a**) IHC 10×: The ciliated respiratory-type epithelium is strongly positive for CK7 (red arrow); (**b**) IHC 5×: The smooth muscle bundles are strongly and diffusely positive for SMA (purple arrow); (**c**) IHC 5×: The nervous tissue is strongly positive for a GFAP-specific staining (green arrow).

## Data Availability

No new data were created or analyzed in this study. Data sharing is not applicable to this article.

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
