# Peer review of "Mature Triphyllic Cystic Teratoma of the Posterior Mediastinum in a Fetus: A Case Report and Literature Review"

_diseases, 2023, doi:10.3390/diseases11040144_

Round 1
Reviewer 1 Report
Title Mature Triphyllic Cystic Teratoma of the posterior mediastinum in a fetus: a case report and literature review
The authors could use more clear language. For example, A teratoma is a neoplasm composed of cell populations or tissues that remind, in their appearance, normal elements derived from at least two embryonic layers. In this sentence, the authors could plainly state that teratomas represent the derivatives of at least two embryonic layers.
The authors state the nature and location of teratomas and their association with congenital abnormalities and that they can cause compression.
Prenatal diagnoses are uncommon and made with ultrasonography; differential diagnosis with other congenital conditions is mandatory. The authors should outline that they refer to the prenatal diagnosis of teratomas. The authors provide a report of a case of teratoma in the posterior mediastinum.
The abstract could also display the key findings, implications and prospects and currently, it is lacking this information.
Intro:
There are multiple cases of capitalising various words for no apparent reason, please comprehensively review this manuscript for language issues.
Teratomas are germ cell tumours that arise from totipotent cells. This implies that this occurs before the formation of the blastocyst is that correct?
More can be added to the developmental aspects of teratomas. For example, the three hypotheses underlying teratoma formation can be expanded on or instead shown in a figure. Since the authors mention that teratomas can be gonadal and non-gonadal, how does this sit with the 3 hypotheses about the origin of teratomas? For example, can the gonadal type arise from hypothesis 3?
Then the authors talk about some stats relating to teratomas, which is useful.
If the authors add a figure to show the three hypotheses, within that figure they can add a section to depict the site of presentation of teratomas as well, this would provide a visual perception to the readers that will help with understanding the rest of the study.
Past analyses show that teratomas are frequently solitary, though, they might sometimes be associated with other malformations and chromosomal abnormalities [5,10]. The authors could give an example of this, please.
The effects of teratoma when present in the chest are also mentioned in addition to diagnoses and treatment. The authors could mention any stats available about the frequency of prenatal and foetal teratomas.
Case reports:
Was there any possibility of the provision of an interpreter to obtain a more detailed medical history from this patient?
Figure 1 is generally informative and the authors have used arrows and other annotations to refer to crucial anatomical structure or the growth within the figure.
The authors offer differential diagnosis and also extensively discuss the relevant steps taken to manage the case including the abortive labour.
The authors could provide a table of abbreviations.
In Figure 2, the authors refer to the anatomical location of the cyst upon dissection and the physical/ macroscopic appearances.
Given the 3 hypotheses regarding the origin of teratomas, which one do the authors see as more likely to be the origin in this case?
In Figure 3, it is interesting that ciliated respiratory epithelium is found on the outer border of the cyst. Since the mainstream epithelial lining of the respiratory tract is the pseudo-ciliated columnar epithelium, where do the authors think this cyst originated from? Also since there is hyaline cartilage, this might narrow it down.
The authors then performed some IHCs to test for various markers.
Since this is a case report and usually lab-based methods and materials may not appear in it, the authors should add methods and a material section to explain how they performed the lab-based techniques (how antibody, what company, what method etc.). Also, it would be good if the authors could add a few sentences and references about the tested markers such as CK7, SMA and GFAP.
Why were these markers tested? And what were the main tissue types found? Please explain this in the main text.
Have the authors tested a proliferation index marker such as KI-67?
Was this teratoma differentiated or undifferentiated and therefore what was the prognosis?
Discussion:
The first paragraph of the discussion has no references.
Of the main types of tissue expected to be seen in a teratoma, which ones were present in this case?
The authors cover in good depth the type of tissue expected to be seen in a teratoma in lines 156-170.
The grade of the teratoma in this study could be mentioned. Also, has the status of p53 expression been established? It is strange that the higher expression of a tumour suppressor could be linked to a worse prognosis in reference 18.
It is interesting that the current teratoma is extracardiac rather than intracardiac, any stats linked to this finding would be useful.
The discussion contains evidence from the literature but the authors have not linked these to the findings of this study. Please state/ recap your results and link them to the literature to be more effective. Therefore, the authors need to slightly shift their narrative to encompass the data obtained and add them to the relevant locations in the discussion. For example, the differential diagnosis stated in lines 206-208 can proceed with the mention of what was conducted in this study to better support it and the same for the rest. The literature sources cited are really good they just need to be better linked to the current study.
I’m not sure what the relevance of CVR is in the discussion. If this was mentioned in the results and was utilised that would have made more sense. Can the authors use this in the results section?
Referral to the current study only appears in sentence 245.
The couple refused extensive testing including cytogenetics.
This paragraph would be more useful in the results:
Histologically, the cystic neoformation was characterized by respiratory epithelium 254 surrounded by smooth muscle with seromucinous glands, hyaline cartilage, and nervous 255 tissue. These features were also confirmed by immunohistochemistry. All said aspects 256 guide us towards a Mature Triphyllic Fetal Cystic Teratoma, grade 0.
Do the authors have any limitations to state or any implications or future prospects to state at the end of the discussion?
Overall, this is an interesting study but the narrative of the discussion needs to be altered plus other suggestions made about other sections.
Some editing is required
Author Response
Dear Editor,
We followed almost all the recommendations of both the reviewers for the revision of the paper (the corrections are evidenced in bold as required). What we couldn’t add are the following:
-We were unable to provide a figure depicting the three etiopathogenetic hypotheses.
-Abbreviations are included in the text and not in a separate table.
- Study limitations and future implications were not included since we thought they are inappropriate for a case report (They are briefly included in the discussion).
-We couldn’t insert scale bars in the figures for technical problems.
We think that in current version the paper could be taken in consideration for publication.
Sincerely
Dr. Cecilia Salzillo
Reviewer 2 Report
This a unique case report combained with literature review important for clinicians.
Is the word “Title” needed in the title?
Please format the Abstract, Introduction and Discussion into a few sections (not containing 1-few sentences) and without listed information
IUGR, cCAML. CK7, TTF1 CK20, GFAP etc. -provide a full name when used for the first time
Histochemistry and Immunohistochemistry methodology is not provided. For the latter provide ab concentrations, a reason to use these abs, negative controls and information how the sections were analyzed
Add scale to Figure2, 3,4 (5x?) add arrows to this figure
Lines 15-154 add references
Provide study limitations
Author Response

(The authors gave the same response as above.)

Round 2
Reviewer 1 Report
Please kindly use a point-by-point response format. I could not find the answer to the comments below:
The abstract could also display the key findings, implications and prospects and currently, it is lacking this information. Done
Intro:
There are multiple cases of capitalising various words for no apparent reason, please comprehensively review this manuscript for language issues.
Teratomas are germ cell tumours that arise from totipotent cells. This implies that this occurs before the formation of the blastocyst is that correct?
More can be added to the developmental aspects of teratomas. For example, the three hypotheses underlying teratoma formation can be expanded on or instead shown in a figure. Since the authors mention that teratomas can be gonadal and non-gonadal, how does this sit with the 3 hypotheses about the origin of teratomas? For example, can the gonadal type arise from hypothesis 3?
If the authors add a figure to show the three hypotheses, within that figure they can add a section to depict the site of presentation of teratomas as well, this would provide a visual perception to the readers that will help with understanding the rest of the study. The authors report they can’t add this to the editor, so if the editor is ok with this, it is fine.
Past analyses show that teratomas are frequently solitary, though, they might sometimes be associated with other malformations and chromosomal abnormalities [5,10]. The authors could give an example of this, please.
The effects of teratoma when present in the chest are also mentioned in addition to diagnoses and treatment. The authors could mention any stats available about the frequency of prenatal and foetal teratomas.
Case report:
Was there any possibility of the provision of an interpreter to obtain a more detailed medical history from this patient? Done
Figure 1 is generally informative and the authors have used arrows and other annotations to refer to crucial anatomical structure or the growth within the figure.
-Abbreviations are included in the text and not in a separate table. The authors report they can’t add this to the editor, so if the editor is ok with this, it is fine.
In Figure 2, the authors refer to the anatomical location of the cyst upon dissection and the physical/ macroscopic appearances.
Given the 3 hypotheses regarding the origin of teratomas, which one do the authors see as more likely to be the origin in this case?
In Figure 3, it is interesting that ciliated respiratory epithelium is found on the outer border of the cyst. Since the mainstream epithelial lining of the respiratory tract is the pseudo-ciliated columnar epithelium, where do the authors think this cyst originated from? Also since there is hyaline cartilage, this might narrow it down.
Have the authors tested a proliferation index marker such as KI-67? Done
Was this teratoma differentiated or undifferentiated and therefore what was the prognosis? Done
The first paragraph of the discussion has no references.
Of the main types of tissue expected to be seen in a teratoma, which ones were present in this case? Done
The grade of the teratoma in this study could be mentioned. Also, has the status of p53 expression been established? It is strange that the higher expression of a tumour suppressor could be linked to a worse prognosis in reference 18. Done
It is interesting that the current teratoma is extracardiac rather than intracardiac, any stats linked to this finding would be useful.
The discussion contains evidence from the literature but the authors have not linked these to the findings of this study. Please state/ recap your results and link them to the literature to be more effective. Therefore, the authors need to slightly shift their narrative to encompass the data obtained and add them to the relevant locations in the discussion. For example, the differential diagnosis stated in lines 206-208 can proceed with the mention of what was conducted in this study to better support it and the same for the rest. The literature sources cited are really good they just need to be better linked to the current study.
I’m not sure what the relevance of CVR is in the discussion. If this was mentioned in the results and was utilised that would have made more sense. Can the authors use this in the results section?
This paragraph would be more useful in the results:
Histologically, the cystic neoformation was characterized by respiratory epithelium 254 surrounded by smooth muscle with seromucinous glands, hyaline cartilage, and nervous 255 tissue. These features were also confirmed by immunohistochemistry. All said aspects 256 guide us towards a Mature Triphyllic Fetal Cystic Teratoma, grade 0. Done
-We couldn’t insert scale bars in the figures for technical problem. The authors report they can’t add this to the editor, so if the editor is ok with this, it is fine.
Do the authors have any limitations to state or any implications or future prospects to state at the end of the discussion? The authors report they can’t add this to the editor, so if the editor is ok with this, it is fine.
Minor language editing
Round 3
Reviewer 1 Report
thanks